# Short communication: Mechanism and Prevention of Irreversible Trapping of Atmospheric He During Mineral Crushing

Stephen E. Cox[1,2], Hayden. B.D. Miller[1,3], Florian Hofmann[1,4], Kenneth A. Farley[1]

[1]Division of Geological and Planetary Sciences, California Institute of Technology, Pasadena, CA 91125, USA
[2]current address: Lamont-Doherty Earth Observatory, Columbia University, Palisades, NY 10964, USA
[3]current address: Earth and Environmental Science Division, Los Alamos National Laboratory, Los Alamos, NM, 87545 USA
[4]current address: Department of Earth Sciences, University of Southern California, Los Angeles, CA 90089, USA

*Correspondence to*: Stephen E. Cox (stephen@stephencoxgeology.com)

**Abstract.** A pervasive challenge in noble gas geochemistry is to ensure that analytical techniques do not modify the
composition of the noble gases in the samples. Noble gases are present in the atmosphere and are used in a number of
manufacturing procedures and by laboratory equipment. Of particular concern is the introduction of atmospheric or laboratory
noble gases to samples during preparation before samples are placed in a vacuum chamber for analysis. Recent work has shown
the potential for contamination of crushed samples with air-derived He that is not released by placing the samples under
vacuum at room temperature. Using pure He gas as a tracer, we show that the act of crushing samples to a fine powder itself
can introduce He contamination, but that this is easily avoided by crushing under liquid or in an inert atmosphere. Because the
He is trapped during crushing, the same concern does not extend to samples that are naturally fine-grained when collected. We
also show model results that demonstrate when this effect might significantly impact samples for (U–Th)/He geochronology
or $^3$He cosmogenic nuclide dating. The degree of He contamination from crushing samples to sizes smaller than the >63 μm
range typically used for geochronology is insignificant for samples at least 1 Ma and 1 ppm U, and the degree of He
contamination from crushing samples to sizes smaller than the 100-500 μm range typically used for cosmogenic nuclide dating
is also insignificant for samples at least 10 ka with typical $^3$He production rates.

## 1 Introduction

The five noble gases are reservoirs of important geochemical information, and the two stable isotopes of He have many uses
for earth scientists. Geochemists measure rare $^3$He (abundance $1.39 \times 10^{-6}$ of total He) as a tracer of tritium in water (Schlosser
et al., 1988), as a cosmogenic nuclide for surface exposure dating (Lal, 1988), as a tracer of mantle fluids (Farley et al., 1995),
and as a synthetic tracer of experimental He degassing in minerals (Shuster and Farley, 2005). The most abundant isotope,
$^4$He, is in turn a measure of temperature-dependent solubility in water (Weiss, 1970), a radiogenic daughter product for
thermochronology (Zeitler et al., 1987), and a signature of dust derived from crustal rocks (Winckler et al., 2005). He gas is
also used as an inert environment in various manufacturing processes and in laboratory cryogenics, sometimes in an
isotopically purified form. There are no radioactive isotopes of He long-lived enough to be found in measurable quantities in

nature, so a complication of these many uses is that, with only two isotopes, the system is underdetermined relative to the number of potential components, and one must use additional contextual information or processing techniques to isolate the desired signal. Because of this omnipresent burden, we read the intriguing results of Protin et al. (2016) with some concern.

Protin et al. (2016) found that fine-grained minerals crushed in the laboratory before measurement appeared to irreversibly adsorb He from the atmosphere, meaning that the He would be adsorbed at room temperature and then retained until the sample was heated to high temperature during the measurement process. The idea that a sample might adsorb He at room temperature would not itself be cause for concern, as it is well known that surfaces can adsorb gas. However, this process is usually assumed to be reversible, meaning that a sample placed in a vacuum chamber will relinquish the adsorbed air as readily as it was first
adsorbed at room temperature or, to speed up the process, under a very mild heating compared to the temperatures used for sample gas extraction (for example, gentle heating to 50–150 ºC prior to extraction at > 1000 ºC is a common technique for cleaning noble gas samples). In contrast, Protin et al. found that the irreversibly trapped He was not released under vacuum during heating in a temperature range distinct from that in which the sample gas was released, which would render impossible distinguishing the two components without isotopic deconvolution.

The problem of trapped air in noble gas measurements is not a new one. In the Ar isotopic system commonly used for K/Ar and $^{40}$Ar/$^{39}$Ar dating, an additional isotope can be used to correct for the pervasive air contamination caused by the high abundance of Ar in the atmosphere (approximately 1%, compared to 5 ppm for He), but this is usually not feasible with He due to the extremely low and relatively uncertain $^3$He/$^4$He ratio of air and the low and variable $^3$He/$^4$He of radiogenic He, and
the potential for mixture with other sources of $^3$He such as mantle-derived He and cosmogenic He. The $^3$He/$^4$He ratio of air is usually assumed to be 1.39x10$^{-6}$, and appears to be invariant in space and time at a level better than the typical uncertainty of He isotope measurements (Lupton and Evans, 2013; Boucher et al., 2018); however, the absolute uncertainty of this ratio is worse than 1% (Blard et al., 2015). Radiogenic He produced directly by the decay of U and Th is 100% $^4$He, but neutrons produced primarily by (α,n) reactions as a result of the same radioactive decay processes produce $^3$He through the reaction
$^6$Li(n,α)$^3$He at a rate dependent on the composition of the mineral, primarily its Li content. Minerals such as apatite (Farley et al., 2001) and zircon (Sliwinski et al., 2018) typically have less than 1 ppm Li, which yields an insignificant quantity of $^3$He, although these too could be compromised by adjacent Li-rich minerals such as micas and amphiboles due to the relatively long ~30 µm stopping distance of the $^3$He produced by the $^6$Li(n,α)$^3$He reaction (Farley et al., 2001). Analytical techniques for (U–Th)/He dating often do not even allow the possibility of measuring $^3$He in the sample, as the isotope is frequently used as a
spike on quadrupole mass spectrometers that are not, in any case, capable of resolving small natural abundances of $^3$He from the ubiquitous HD background. Finally, an isotopic correction for air contamination would have to assume no isotopic fractionation during the trapping process and a well-known laboratory air composition, but the extremely small structures that would be required to irreversibly trap He within the surface layer of minerals would be likely to fractionate the gas isotopically,

and the pervasive use of He and other compressed gases in laboratories makes the composition of the laboratory atmosphere variable and difficult to predict without frequent measurement.

Typical analytical procedures for (U–Th)/He dating, cosmogenic $^3$He dating, and other He isotope analyses call for approximately sand-sized particles with dimensions in the range of 50–500 μm. There are several reasons for this. In the case of (U–Th)/He analyses, the stopping distance of α particles in minerals is 15–40 μm, depending on the mineral and the energy of the particle. Because α particles that travel outside of a mineral after emission during U and Th decay are lost even as the radioactive parents remain, one must make an alpha ejection correction in the course of (U–Th)/He dating, and the magnitude and uncertainty of this correction becomes very large as the grain size analyzed approaches the ejection lengthscale (Farley et al., 1996). In the case of cosmogenic $^3$He analyses, crystals larger than 500 μm can suffer from contamination with magmatic He in fluid inclusions, and crystals smaller than 100 μm can be more contaminated by atmospheric He (Blard, 2021). Practically, powders finer than sand-sized are difficult to handle in the vacuum system, so one tends to choose coarser material when possible. It is also simpler to ensure the mineralogical purity of the samples when they are large enough to manipulate easily and observe under a microscope, but not so large that it is impossible to isolate individual crystals or to see inclusions. There are two reasons one might choose to analyze minerals that are finer than sand-sized: in some cases, the grain size of the sample may simply be smaller; in other cases, one might wish to crush a sample to avoid pervasive small inclusions or intergrowths, or one might crush samples as part of the analytical process, for example to separate magmatic and cosmogenic endmembers in cosmogenic $^3$He dating (Kurz et al., 1986). Typically, one can isolate mineralogically pure samples that are no smaller than 50 μm, which is above the size range of concern based on the results of Protin et al. (2016), and low-He samples crushed to a powder in vacuum before fusion can be handled in a way that minimizes atmospheric He exposure. However, there are many cases in which the material is naturally fine-grained. A couple of examples include polycrystalline iron oxides, which can be dated using the (U–Th)/He and (U–Th)/Ne methods, and which comprise a range of crystallites as small as less than 1 μm (Farley and Flowers, 2012; Farley and McKeon, 2015), interplanetary dust particles, in which $^3$He is measured as a constant flux proxy, and for which the size range of interest is a few μm to less than 35 μm (Farley et al., 1997), and cosmogenic $^3$He studies in which finer material is deliberately targeted in order to avoid the fluid inclusions that are more prevalent in larger crystals (Puchol et al., 2017; Blard, 2021).

The study of Protin et al. (2016) used isotopic deconvolution and sequential crushing in an attempt to quantify the amount of He contamination from laboratory air. Our approach is much simpler: while we sought to confirm in broad terms the results of Protin et al., we do not attempt to quantify the surface area effect that they document. Rather, we accept the premise that contamination from atmospheric He is possible and instead take advantage of pure He gas (200,000 times more concentrated than He in the atmosphere) to investigate the mechanism of contamination using a coarser size fraction and to arrive at a solution that allows us to avoid the problem in practical situations. While we do not attempt to degas the samples before

crushing or to quantify the isotopic composition of the He components, the use of pure He gas during crushing makes these tests very sensitive compared to the concentrations of a few ppm that one would typically find in a laboratory atmosphere.

## 2 Methodology

### 2.1 Sample Preparation

We prepared samples in a variety of ways to simulate different laboratory noble gas contamination scenarios. In each experiment, we crushed samples of San Carlos olivine in the presence of different gas mixtures at atmospheric pressure, and in some cases under fluids. Initial sample preparation involved picking inclusion-free San Carlos olivine from working samples in the Caltech Geological and Planetary Sciences mineral collection. The San Carlos olivine standard has a variable but moderate He concentration of less than about $7.5 \times 10^{-12}$ mol $g^{-1}$ (Mohopatra and Murty, 2000). We chose the San Carlos standard because inclusion-free crystals are readily available and because $7.5 \times 10^{-12}$ mol $g^{-1}$ is small compared to the contamination we expect in a pure He environment based on Protin et al. (2016). The smallest degree of contamination the authors observed in the 50–125 µm size fraction was $8.2 \times 10^{-14}$ mol $g^{-1}$ from crushing in atmospheric He, which we scale to $1.6 \times 10^{-8}$ mol $g^{-1}$ in the 200,000 times more abundant pure He used in this study. Indeed, the He concentration in San Carlos olivine is more than 150 times less than the highest concentrations measured in this study. We treat He concentrations less than $1.5 \times 10^{-11}$ mol $g^{-1}$, or twice the highest concentration recorded by Mahopatra and Murty (2000), as the background level for this study because of the use of non-degassed olivine in the experiments (see background boxes on Figures 1 and 2).

After initial picking, we separated aliquots of the sample for the different crushing treatments. The crushing device we chose was a standard agate mortar and pestle, which we cleaned extensively between experiments using a scouring pad, water, and isopropyl alcohol. With the exception of crushing performed in the presence of air, for which we simply crushed the samples on the bench in a laboratory in which no compressed gas cylinders are used or stored, we created the gas atmospheres by placing the mortar and pestle in a sealed plastic zipper bag during crushing. The gas was introduced through a small opening in the zipper bag, such that the bag inflated and filled with the gas but was not sealed sufficiently well to increase the pressure significantly about ambient atmospheric pressure.

For the initial set of experiments, we simply took the finest portion of the powder to check for He contamination in the different aliquots. After finding initial evidence of contamination, we sieved the samples at 50 µm and measured the finer fraction to confirm that the surface area of the crystals was an important control on the degree of He contamination. Finally, we repeated the initial experiments, and added some additional tests, in order to determine whether crushing itself or fine grain size was the controlling factor on He contamination. Additional experiments included crushing in vacuum and in a He-free atmosphere and then later exposing to the 100% He atmosphere, crushing under water in the presence of a 100% He atmosphere, and leaching the sample for 20 minutes in concentrated nitric acid after crushing in the presence of a 100% He atmosphere. We

sieved all samples in the second round to a grain size of 37-50 μm after crushing in order to improve the uniformity of the potentially-contaminated surface area in the measured samples.

## 2.2 Mass Spectrometry

We measured the first round of samples using laser degassing followed by measurement with a Pfeiffer Prisma quadrupole used routinely for (U–Th)/He dating at Caltech (House et al., 2000). Measurements were interspersed with background analyses and standardized using a calibrated $^4$He reference standard and a $^3$He spike measured against the samples and the standard. The $^4$He chamber background was less than $5 \times 10^{-14}$ mol, the $^3$He spike was $1.3 \times 10^{-13}$ mol, and the $^4$He reference standard was $3.2 \times 10^{-13}$ mol. For this set of experiments, we heated each sample to at least 800 ºC to verify that reversibly adsorbed He, or even He that could be separated from the sample through step heating, was fully removed. We then heated the samples to about 1500 ºC to fully degas them; any additional He in these measurements represents irreversibly trapped He.

After verifying that the He trapping was irreversible, as reported by Protin et al. (2016), we conducted the second set of experiments using a single high temperature extraction with a vacuum furnace to allow for larger samples, and we measured the trapped He using an MAP 215-50 mass spectrometer. Measurements were interspersed with background analyses and standardized by bracketing with a calibrated $^4$He-doped air reference standard ("Caltech Air"). Procedural blanks ranged from $1.4 \times 10^{-14} \pm 4 \times 10^{-16}$ to $6.6 \times 10^{-14} \pm 3 \times 10^{-16}$ mol and were mostly dependent on the cleanliness of the vacuum furnace; blank/signal ratios ranged from 0.9% for the samples crushed in pure He to 73% for the sample crushed under water. The $^4$He concentration of the air standard was $3.6 \times 10^{-12}$ mol. Extractions were performed at about 1500 ºC to fully degas the samples.

## 3 Results and Discussion

### 3.1 He contamination in crushed samples

As expected based on the results of Protin et al. (2016), we found significant He contamination in samples crushed in the presence of pure He (Figures 1 and 2, appendix tables S1 and S2). The contamination was retained at surprisingly high temperature, with more than 20% of the additional He retained even after pumping the samples under ultra-high vacuum for several hours and then heating to more than 800 ºC for ten minutes (Figure 1). Because the crushing was carried out at laboratory temperatures (~21 ºC), this result suggests that the trapping of He is indeed irreversible. He merely adsorbed onto the surface of the crushed olivine would be released after placing the samples under vacuum, and certainly after even mild heating. We suggest two possible reasons for this anomalous irreversible He trapping and retention. One, as suggested by Protin et al. (2016), fine-grained samples may exhibit a "lobster pot" trap for He. This analogy implied that the He is trapped by some mechanism that does not allow it to escape (like a lobster pot, which lobsters can easily enter but then not escape due to the geometry of the trap). Under this analogy, He exposure at any time after the sample was crushed would cause contamination, and naturally fine-grained samples would be subject to contamination from the environment. On the other hand,

the violent crushing action itself may cause the He to be trapped, perhaps due to tiny fractures or other structures that open during crushing and then heal or anneal immediately. In this case, the He would only be trapped during crushing, but would not be susceptible to irreversible He contamination after crushing, and naturally fine-grained samples would not be subject to the same contamination. Our second set of experiments was designed to distinguish between these two scenarios.

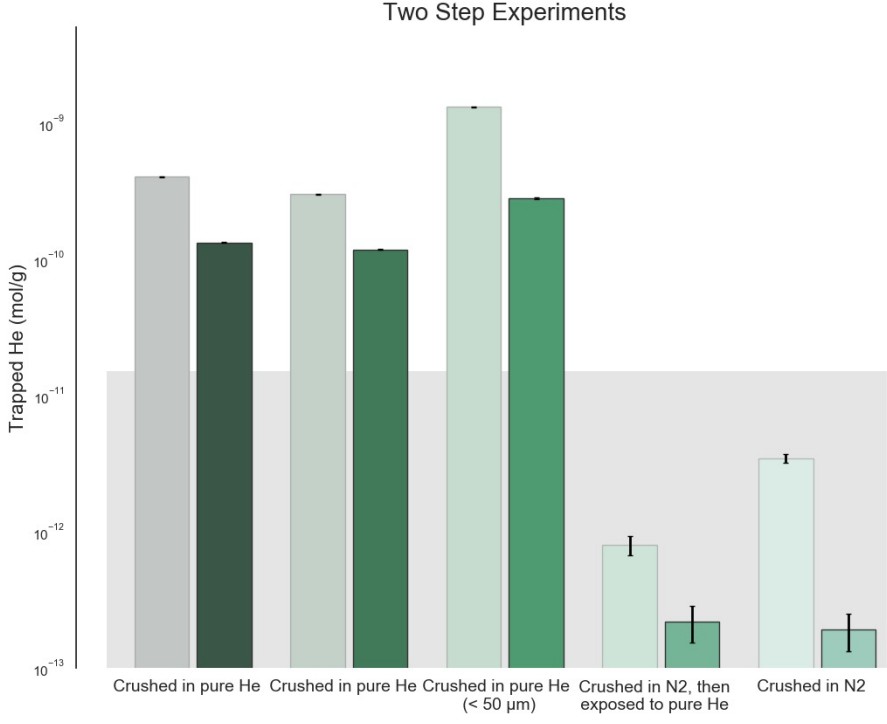

**Figure 1: He trapped by crushing process in initial two-heating-step experiments. The orange background rectangle shows the approximate background level of the experiment based on the variable natural He content in San Carlos Olivine (Mohopatra and Murty, 2000; this work). Analytical standard errors shown are 2-σ and account for QMS counting statistics, and are small compared to both the natural background and the He enrichment measured in these experiments. The larger pale boxes show the initial low-**
170 **temperature (800-900 ºC) extractions of each sample, while the foreground boxes show the second high-temperature (1500 ºC) extractions. Compared to the samples that were either crushed in pure $N_2$ and then soaked in He or simply crushed in $N_2$ and then analyzed, the samples crushed in the presence of He exhibit a substantial contamination from the He atmosphere during crushing. The aliquot sieved to less than 50 μm shows an even greater He enrichment, consistent with a surface area dependence as shown by Protin et al. (2016).**

**3.2 Irreversible He trapping due to crushing, and trapping of other noble gases**

The second set of experiments confirms that the irreversible He trapping occurs during the crushing process rather than a result of the grain size of the samples, or as a result of the fact that they have been crushed. We show dramatic reductions in He contamination, to levels similar to the detection limit of this methodology, in several different scenarios (Figure 2). First, crushing in a neutral atmosphere ($N_2$) and then immediately exposing to pure He results in virtually no additional He
contamination. This shows that there is not an anomalous He-specific trapping mechanism that remains accessible after the

crushing event, but it does not rule out an unexpected irreversible surface trapping mechanism for any gas due to crushing or grain size. In another experiment, however, we crushed in high vacuum using a magnetically-actuated high vacuum crusher, then immediately flooded the vacuum system with pure He. In this case, no other gas was present in the volume of the chamber during crushing, and pure He was the first gas exposed to the sample after crushing, but the experiment still showed far less

He trapping (about 5% of the case in which the sample was crushed in the presence of pure He). Finally, we also crushed samples in the presence of a mixture of 25% each of He, Ne, Ar, and Kr. In this case, the sample trapped approximately 25% as much He as the sample crushed in the presence of pure He. This final experiment, combined with the evidence that pure He is not trapped after either vacuum crushing or crushing in an $N_2$ atmosphere, is consistent with a model in which the sample traps gases present in the atmosphere during crushing according to their volume ratio, rather than trapping He through a

mechanism that excludes other gases.

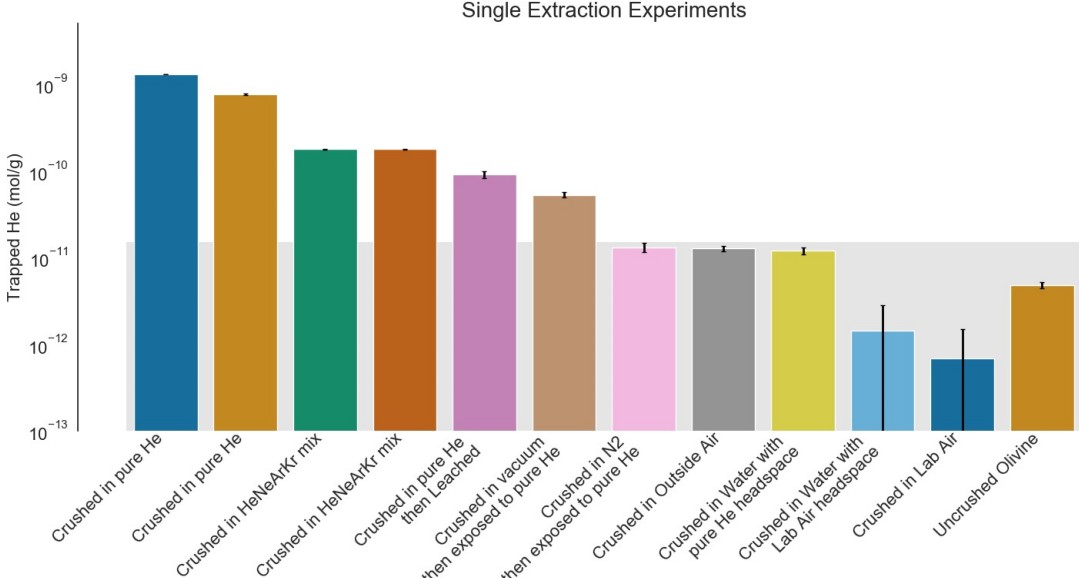

**Figure 2: He trapped by crushing process in single-step follow-up experiments. The grey background rectangle shows the approximate background level of the experiment based on the variable natural He content in San Carlos Olivine (Mohopatra and Murty, 2000; this work). All samples were sieved to a size range of 37-50 µm to reduce variability from sampling. Analytical standard**
**errors shown are 2-σ and are small compared to both the natural background and the He enrichment measured in these experiments. The samples crushed in the presence of pure or enriched He show significant enrichments in He. Acid leaching reduces the amount of He contamination, but not to background levels. The sample crushed under vacuum then immediately exposed to pure He also shows a He enrichment, but it is only a few percent of the enrichment seen in the samples exposed to He during crushing. Samples crushed in a neutral atmosphere and then exposed to He, and the sample crushed under water with a pure He atmosphere, are**
**indistinguishable from background.**

### 3.3 Mechanism of trapping during crushing

The mechanism of trapping during crushing is a matter of speculation, but it is important in that it is clear from our results and from earlier work by Niedermann and Eugster (1992) and Protin et al. (2016) that the trapping is irreversible. The trapping appears to apply to heavier noble gases as well (Niedermann and Eugster, 1992), meaning that it is not enabled by a mechanism

dependent on the unusual mobility and small size of He. It also appears to cease shortly after the physical impact that causes it to occur, as evidenced by the experiments showing minimal contamination of sample crushed in a He-free atmosphere or under vacuum and then later exposed to pure He. Therefore, we consider that the most likely mechanism is that microfractures form during crushing and then close or anneal shortly thereafter. This mechanism would be consistent with the findings presented here and would also suggest that naturally fine-grained samples are not subject to the same phenomenon.

## 210   3.4 Implications for experimentalists

    While potentially troublesome for experiments that require crushing of low-He samples to a very fine grain size in the laboratory, these results are not nearly as concerning as the possibility that all fine-grained samples trap He, or even the possibility that crushed samples might trap He at any point after crushing. Because the problem is largely restricted to the time of crushing itself, we explored several solutions that exclude He from this part of the sample preparation. As mentioned above,

crushing in a neutral ($N_2$) atmosphere then exposing to He did not result in significant contamination, so using a neutral atmosphere such as $N_2$, which is relatively cheap as both a compressed gas or a product of liquid nitrogen boiling, is a solution to the problem (Figure 2). Indeed, in most cases of typical He concentrations in samples, crushing in air (which is only 5 ppm He) and taking care to avoid unusually concentrated or isotopically fractionated sources of He in the laboratory is adequate, but extra care should be taken in particular for very young cosmogenic [3]He samples (Blard et al., 2006).

    The other solutions we explored were crushing under water and crushing then acid leaching the sample. Leaching the sample in concentrated nitric acid for twenty minutes removed about 45% of the mass of the 37-50 μm fraction and removed about 90% of the He (Figure 2). While this is a dramatic improvement and shows that trapping occurs mostly on the outer edges of the fragments created during crushing, it is not as effective as simply avoiding He through crushing in a neutral atmosphere.

It is also not as effective as crushing under water, which may in many cases be the simplest solution available. He solubility in water is quite low compared to other gases (Weiss, 1970), so working under water is a good way to exclude He, and crushing under water is often preferable anyway because it helps prevent sample loss. For this test, we flooded a bag with pure He in the same way as the experiments showing dramatic increases in sample He concentration during crushing, but also filled the mortar with a ~1 cm layer of tap water before crushing the sample. The level of He contamination was below the detection

limit of this methodology, which is limited by the natural background of He in the San Carlos olivine (Mahopatra and Murty, 2000), and similar to the sample crushed in air despite the presence of 200,000 times the partial pressure of He in the atmosphere just above the water layer (Figure 2).

    In addition to the imperative to avoid unwanted sample contamination, the irreversible trapping mechanism we observe may

actually be useful for applications in which trapping synthetic noble gas mixtures in minerals would be desirable, such as mineral standard preparation or isotopic spiking in environments such as on planetary bodies. The scope of the current work does not explore the degree or variability of isotopic fractionation during trapping or the extent to which the degree of

contamination could be controlled by different crushing techniques or more careful size separation, both of which would be critical for such applications.

## 4 Modeling of impact of He contamination for thermochronology

### 4.1 Model description

In order to conceptualize the observed phenomena and to make predictions about the effect of anomalous irreversible He trapping on mineral samples used for geochronology, we constructed a simple geometric model to represent one physical explanation consistent with our observations. In this model, presented in the appendix in a Jupyter notebook, the amount of He contamination due to trapping decreases exponentially from the surface of the crystal. Crystals are modeled as a sphere, with He contamination entering from the outer surface. The crystal is approximated as spherical shells of 1 μm thickness, with a per-unit-volume He contamination exponential rate constant of 0.29 $\mu m^{-1}$, which is based on the ~90% of contamination removed by leaching ~45% of the mass of the 37-50 μm samples used here. The choice of an exponential model for the contamination is a hypothesis designed to be consistent with the observation that removing about half of the sample by leaching removed most but not all of the He contamination. If the phenomenon were merely surface contamination, leaching would have removed all of the it. Conversely, if it were pervasive in the mineral, leaching would only remove a similar proportion of the contamination. While a simple surface area-dependent contamination model would account for the grain size dependence of He contamination, a model in which contamination decreases exponentially with depth into the crystal surface makes it possible to account for the observation that the majority of the contamination occurs near the surface but some extends deeper into the crystal.

For the purpose of comparison to He concentrations in geo/thermochronology, we simulate the radiogenic ingrowth of He in apatite crystals of several age and U concentrations. Alpha ejection is ignored, so there is no difference between He from U and Th decay, and so only U is considered for the sake of model simplicity. The same principles can be applied to radiogenic $^4$He and cosmogenic $^3$He in other minerals. In the case of cosmogenic $^3$He measurements, the greatest concern will be the effect of $^4$He contamination on the magmatic $^3$He correction (Protin et al., 2016; Blard, 2021). While $^3$He contamination is also a possibility, this effect will be scaled down by the $^3$He/$^4$He ratio of the contaminant, which is $1.39 \times 10^{-6}$ if the contaminant is atmospheric He.

This model allows us to scale the degree of contamination to different grain sizes, and to compare the atmospheric He contamination to the radiogenic He concentration of minerals of different ages and U concentrations. The reference point used for these scaling calculations is a crystal with a diameter of 45 μm (radius 22.5 μm), in the middle of the sieve range used for the second round of experiments, and an amount of He contamination equal to $1.37 \times 10^{-14}$ mol $g^{-1}$, which represents two times the worst contamination observed in our pure He-soaked experiments ($1.31 \times 10^{-9}$ mol $g^{-1}$), scaled down to 5.24 ppm (the

atmospheric concentration of He), both to present a conservative model and to account for the possibility of slightly elevated $^4$He in laboratory environments due to the use of concentrated He in laboratory processes.

The concentration of the outer shell of the r = 22.5 μm model is calculated by creating a series of shells assuming spherical geometry, meaning that the volumes are simply given by equation 1. The value $V_n$ is the volume of the shell beginning at $r_n$, where $r_n$ is the outer radius of the nth shell, counting from the outside of the crystal inward, and $r_{n+1}$ is the inner radius of shell n, or the outer radius of shell n+1. The fractional volume of the shell is then given by equation 2.

The value $V_{n,frac}$ is the fractional volume of shell n, $V_n$ is the volume of shell n, $r_0$ is the radius of the crystal, and a is the width of shells used in the model (a = $r_n$ – $r_{n-1}$). Note that a = 1 μm except for crystals <10 μm in radius, in which case we split the crystal into ten shells. We then use this fractional volume to weight the exponential fit of the He contamination for each shell by volume, an effect which becomes more significant for smaller crystals in which the outer shell volumes differ dramatically. The relative per-unit-volume He contamination is given by equation 3.

The value $[He]_{rel}^n$ is the relative He contamination of shell n. Note that for the outer shell (n=0), $r_n$ = $r_0$, so the He contamination is defined relative to the outer shell. Based on our measurements, we assume that the total concentration of contamination He in the crystal will be $1.37 \times 10^{-14}$ mol g$^{-1}$, then we determine the fraction of this that is present in the outer shell. We used olivine for the experiments as described in Section 2.1 but we use apatite for these calculations in order to demonstrate the relative significance of the contamination in the mineral most commonly used for He thermochronology. The volume concentration of He contamination in the outermost shell is given by equation 4.

The value $[He]_T = 1.37 \times 10^{-14}$ mol g$^{-1}$ and $\rho_c$ is the density of the crystal. We note that while the density of different minerals is small compared to the likely variability of the cracking effect, we are also assuming that the geometric model of cracking applies similarly to different crystal systems. The goal of this work is to show the order of magnitude of the problem and suggest ways to avoid it when it may be significant, but if this model were to be used quantitatively, more careful investigation of the behavior of individual crystal systems might be warranted.

We then take the volume concentration of He in the outer shell $[He]_{outer}$ and apply it to the outer shell of modeled crystals of different sizes in order to calculate the relative contamination in each. For each test radius $r_{test}$, we break the crystal up into either $r_{test}$/(1 μm) shells of a=1 μm width or, in the case of crystals <10 μm in radius, 10 shells of a=$r_{test}$/10 width. We then calculate relative He concentrations in each shell in the same manner described above, then calculate the He concentration in each shell n using the equation 5.

The outer shell concentration $[He]_{outer}$ will simply be the value calculated for the 45 μm crystal, weighted according to the actual volume of the test crystal outer shell, and the inner shells will be weighted by both volume and the exponentially

decreasing cracking frequently defined above. The total amount of He contamination is then given by simply summing over all the shells n (equation 6).

We then compare the total contamination $[He]_{total}$ for a given crushed crystal of size $r_{test}$ to predicted He concentrations for crystals of the same volume and a given age and U concentration to show for which (U–Th)/He samples the effect might be

important, or a given age and ³He production rate to show for which cosmogenic ³He samples the effect might be important. The same logic can be applied to the direct effect of ³He contamination on cosmogenic ³He by simply considering the He concentrations that result from these calculations and the dramatically lower ³He concentration (the ³He/⁴He of atmosphere, $R_a = 1.39 \times 10^{-6}$), but because of this difference in isotopic abundance, the effect of atmospheric ⁴He contamination on the magmatic ³He correction applied to cosmogenic measurements is larger.

$$[\textbf{1}] \quad V_n = \frac{4}{3}(r_n^3 - r_{n+1}^3)$$

$$[\textbf{2}] \quad V_{n,\text{frac}} = V_n \Big/ \sum_{i=0}^{r_0/a} V_i$$

$$[\textbf{3}] \quad [He]_{rel}^n = e^{0.29(r_n - r_0)}$$

$$[\textbf{4}] \quad [He]_{outer} = [He]_T\, \rho_c V_{0,\text{frac}} \frac{[He]_{rel}^0}{\sum_{i=0}^{r_0/a} [He]_{rel}^i}$$

$$[\textbf{5}] \quad [He]^n = [He]_{outer}\, [He]_{rel}^n\, V_n$$

$$[\textbf{6}] \quad [He]_{total} = \sum_{i=o}^{r_{test}/a} [He]^i$$

**4.2 Model results for different crystal sizes and radiogenic He concentrations**

Assuming no loss to diffusion or ejection of alpha particles during radioactive decay, radiogenic He grows into crystals at a rate proportional to the concentration of ²³⁸U, ²³⁵U, ²³²Th, and other minor nuclides that decay by alpha emission. The model allows us to compare the He contamination observed here to radiogenic He concentrations in minerals of a given U concentration (the model applies to other alpha emitters, the concentrations of which can simply be scaled according to alpha activity) and age, or to cosmogenic ³He concentrations in minerals subjected to a given cosmogenic ³He production rate and

age, and to adjust the level of contamination according to the geometric model of He trapping that assumes exponential decay

of the degree of He contamination with depth into the crystal. This simple model reflects just one hypothesis for the nature of He trapping that is consistent with our observations, but it provides an order-of-magnitude tool for determining under which situations even the relatively simple precautions suggested in Section 3.4 need to be considered.

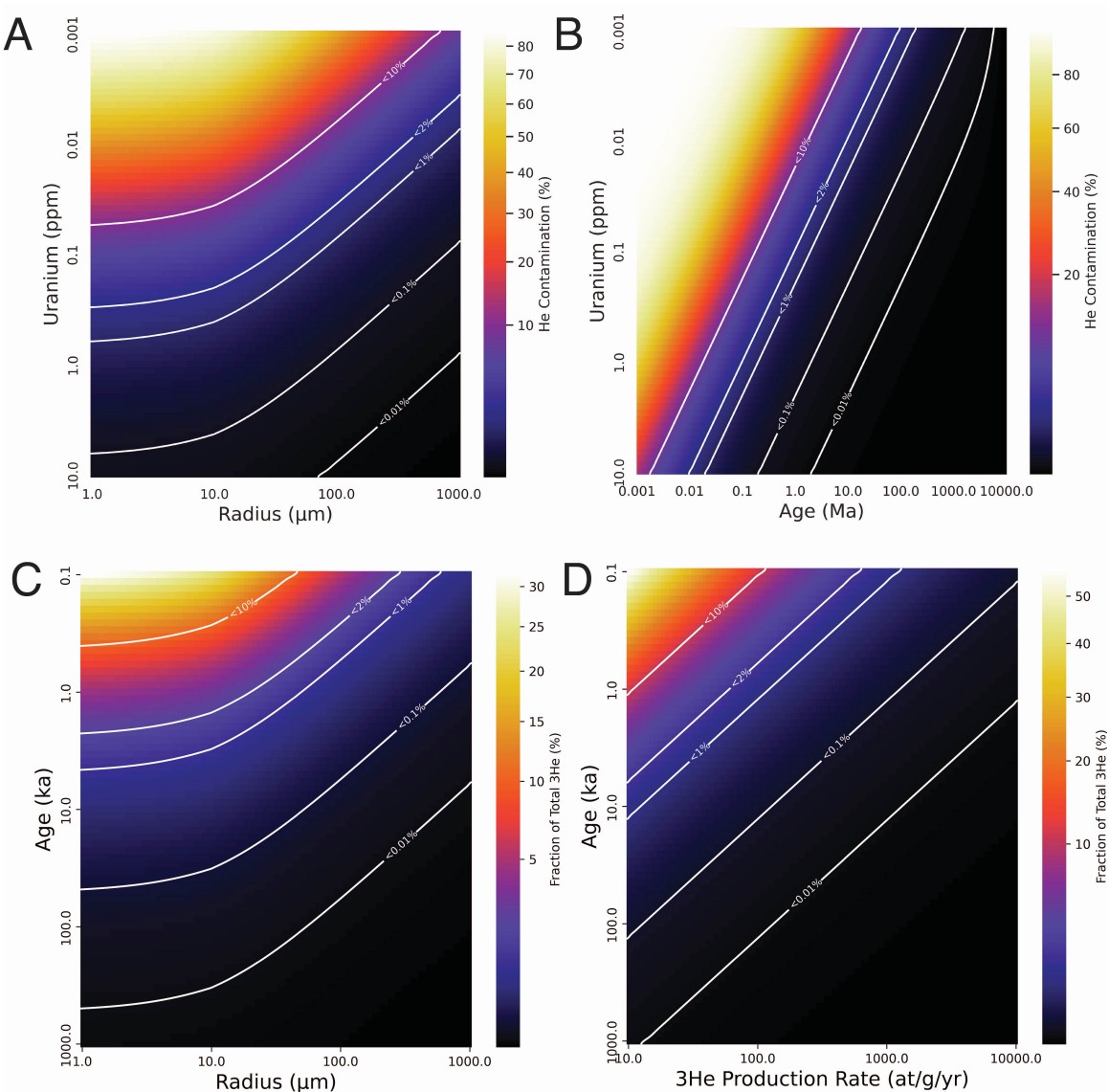

**Figure 3: Modeled effects of He contamination on radiogenic (U–Th)/He apatite (panels A and B) and cosmogenic $^3$He olivine (panels C and D) samples. Panels A and B show contamination as a fraction of total He (radiogenic and contamination) due to cracking in crystal fragments of a certain size, age, and U concentration. Panel A shows contamination as a function of crystal radius and U concentration with a fixed age of 1 Ma. Panel B shows contamination as a function of age and U concentration with a fixed radius of 31.5 μm (reflecting a 63 μm sieve size). Panels C and D show the fraction of total cosmogenic He represented by the correction that would be applied to a cosmogenic $^3$He measurement if the contamination $^4$He is erroneously assumed to be magmatic $^4$He with a ratio of 8 $R_a$ (following Blard, 2021). The degree to which this correction is incorrect will be a function of the actual difference between the $^3$He/$^4$He ratio of the contamination He and the assumed or measured magmatic He. Panel C shows fraction of total $^3$He**

as a function of crystal radius and age assuming a fixed 3He production rate of 125 at/g/yr. Panel D shows fraction of total $^3$He as a function of production rate and age assuming a fixed crystal radius of 50 μm (representing a minimum recommended cosmogenic sample size of 100 μm, following Blard (2021). The spherical model used here is adequate as an approximation for most crystals, but the model code can be modified to account for extreme geometries. Only very young or low-U radiogenic samples and very young or low $^3$He cosmogenic samples will exhibit a problematic degree of He contamination compared to radiogenic He ingrowth.

A selection of model results is presented in Figure 3. The model results for radiogenic samples (panels A and B) show that while He contamination represents a very high fraction (>20%) of the radiogenic He in small crystals (< 45 μm) of moderate age and very low U concentration (1 Ma and 10 ppb) or moderate U concentration and extremely young age (10 ka and 1 ppm), the fraction quickly becomes small (<5%) for samples in the range of (U–Th)/He applications (1 ppm U, 1 Ma, at least 10 μm radius) and is insignificant for typical samples (for example, <0.2% for a 10 ppm U, 1 Ma mineral with a radius of 63 355 μm). This means that while rock crushing for separation of accessory minerals like apatite is typically performed using techniques like disk milling that would be difficult to adapt in a way that keeps the mineral under water, the He contamination problem is nonetheless insignificant for typical samples used for (U–Th)/He. Only exceptional samples with very small, young, and low U concentrations will require special care to avoid He contamination during processing. Similarly, the model results for cosmogenic samples (panels C and D) show that the effect of the contamination He on the magmatic correction for $^3$He 360 will be small for most samples >10 ka with typical cosmogenic $^3$He production rates, so the possibility of atmospheric contamination during sample processing most only be accounted for in samples of exceptionally small size, age, or with unusually low cosmogenic $^3$He production rates.

## 5 Conclusions

These results confirm the experimental data of Protin et al. (2016). However, we find overwhelming evidence that the 365 irreversible trapping of He occurs during the crushing process rather than as a result of the grain size of the sample. The effect will still scale with grain surface area, as it appears that the trapping happens in the outer layer of the crushed fragments and can be mostly removed by acid leaching. However, one must only avoid He during crushing in order to avoid contamination of geological samples. Naturally fine-grained samples and samples crushed in the absence of He do not exhibit the same effect. We hypothesize that the crushing process opens small damage zones that are quickly healed or annealed, and that can only 370 irreversibly trap gas that is exposed to the sample before the healing occurs. Our experiments demonstrate that this process is complete within a few minutes. The simplest means of avoiding contamination is to crush samples under water or other liquid, so we recommend crushing under water for all low-He samples that must be crushed to a fine powder in the laboratory. This solution will work for any mineral, and for both isotopes of He and for other noble gas species. Samples of the grain size commonly used for most measurements, such as (U–Th)/He dating and cosmogenic $^3$He dating, and naturally fine-grained 375 samples are not susceptible to this problem.

**Data and code availability**

The code and data used in this study are included in the appendix.

**Author contributions**

SC, HM, FH, and KF contributed to conceptualization of the study. SC and HM designed and executed the experiments. KF advised on laboratory analyses and experimental design. SC, HM, FH, and KF participated in discussion about trapping mechanism and designed adjustments to common laboratory procedures. SC created the software model, prepared figures, and prepared the original draft. HM conducted model validation. SC, HM, FH, and KF contributed to editing of the manuscript.

**Competing interests**

The authors declare no competing interests.

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
