# Peer review of "Short communication: Mechanism and Prevention of Irreversible Trapping of Atmospheric He During Mineral Crushing"

_Geochronology, 2021_

## Author Response (AR1)

**Response to Reviewer 1**

The short communication "Mechanism and Prevention of Irreversible Trapping of Atmospheric He During Mineral Crushing" aims to show through experiments and a simple model that while contamination of samples by exogenous He sources is possible during sample preparation, it is either relatively inconsequential or easily avoided in most cases. I am convinced by their experiments, and support its publication, however if possible, I suggest a few additional experiments that could test the proposed physical mechanism of He trapping. The paper is well-written and easy to follow (with one exception noted below), although I think some editing of the figures can help the authors make the case even more clearly.

Additional experiments:

The authors speculate that He (or other gases) is trapped within quickly annealing cracks, rather than adsorbed onto fine-grained surfaces. This makes sense and fits most of their data. It does not completely explain why the "Vacuum crushed then He soaked" experiment is more contaminated than the "N2 then He soaked" experiment, although this could be due either to the N2 saturating the cracks prior to introduction of the He or the timing of He introduction was inconsistent, and it just happened to be introduced later in the N2 experiment, after the cracks had fully healed. (This might be useful to point out on its own.) I was surprised that the vaccuum crushed experiments showed some contamination, but this observation allows for a test of the contamination mechanism. The authors could repeat the vacuum crushing experiment but vary the time of He introduction (perhaps ranging from seconds to several minutes or longer after crushing). They should see that the amount of contamination decreases after waiting longer. If possible to perform, I think this "time-series" could help elucidate the trapping mechanism. I don't think these additional experiments would be necessary for publication, but would enhance the paper's completeness.

Presentation of the figures:

In Figures 1 & 2, the experiment names are sometimes confusing. It should be clearly specified for each experiment (1) what condition the sample was crushed under and (2) what treatment after crushing was done (soaking, leaching, seiving, etc). This order of operations is important to the conclusions but not clear in some of the labels. For example, "N2 then He Soaked"--this was crushed under N2 and then soaked in He, but the label could be read as crushed first, then soaked in N2, then soaked in He. Another example, is "Lab Air" a sample crushed in air, or is it a measurement of lab air? Presumably the former.

Also in Figures 1 & 2, the "background level" color should be consistent (it's tan in 1 and grey in 2), and could be labeled on the figure. But this also raises the question, how was this background level determined? Some of the experiments have less He than this, so it's not really a "background". Does it even need to be included?

In Figure 1, I found the "foreground" and "background" an odd and confusing way of showing the two temperature steps. They could instead be shown simply by either stacked or adjacent bars.

Figure 3 is very hard to follow. The modeling exercise is useful, with caveats about the quantitative applicability clearly explained, and broad conclusions interesting and relevant. But the figure is too hard to read. If the authors wish to present the model calculations this way, at the very least, either the line styles should be more systematic (e.g. all 10 ppm U calculations in blue, all 0.01 ppm U calculations in red, or something like that) or their order in the legend should match the order in the plot so they can be more easily referenced. But I think there is an easier way to present the data. The crystal radius is not really a variable of interest because the way the model is set up, larger crystals would necessarily have less contamination, so it's kind of a waste of an axis. The variables of more interest to the conclusion are age and U concentration, so it would be more useful to have one of those as the x-axis. Here is just one idea off the top of my head that I think would make the results more intuitive: a two-panel figure where one panel is calculations for, say, 10 micron radius and the other for 100 or 1000. The x-axis is age and the y-axis is He contamination %. Then plot the calculation as points, using a consistent symbol for the various U concentrations, say triangles for 0.01 ppm, squares for 0.1 ppm, and circles for 10 ppm. Then it is easily demonstrated that contamination is only important for young and/or low U samples, and one does not need to hunt around the figure to find that information. If the authors are so inclined, perhaps the best way to present the results is as a contour or density plot, where the x and y axes are age and U concentration, and the plotted value is calculated contamination %; again doing two panels of this, one for small grains and one for large grains.

Conclusions:

The sentence "Our experiments demonstrate that this process is complete within a few minutes, and it seems likely that the entire process occurs on the sub-µs timescale of the propagation of pressure waves through the minerals" is not supported by the experiments, as there are no hard quantitative constraints on the timescale of the processes occurring (without doing the time series I mentioned above). Furthermore, the sentence is confusing. Does the process take minutes or microseconds? Plus the mention of pressure wave propogation hasn't been mentioned before as a process. Therefore this is not a conclusion. This sentence needs some revision. Otherwise, the writing is clear and organized.

We thank the reviewer for thoughtful and encouraging comments. We agree with the substance of the review, although logistical concerns limit our ability to expand the analytical work. We describe our plans to address the reviewer's comments in detail below.

Regarding the additional experiments, it is not logistically feasible for us to perform these quickly at this time due to other obligations and the fact that most of us have changed

institutions. We intended the "vacuum then He soaked" experiments as a test of how much He was adsorbed during crushing rather than by less time-sensitive, probably reversible processes after crushing, and we will add additional discussion to the text in light of both this reviewer's comments and the extensive comments provided by reviewer 2 on the same subject. We believe that we can adequately address these questions in discussion and by reference to previously published data (as pointed out by reviewer 2) without the need for additional measurements.

The modifications suggested to figures 1 and 2 make sense. See the new versions attached. We will also clarify in the text that the plotted "background level" is better described as the detection limit of the method and is conservatively high as a result.

While it is true that in figure 3 the degree of helium contamination is a simple function of grain radius, the same is true of uranium concentration and age as well, and those two properties are interchangeable. We presented the results according to grain size because this is the variable most easily controlled by the person selecting samples for analysis, so we want to preserve this view. In acknowledgement of the reviewer's concerns and the original intention to present a way to quickly identify the minimum acceptable grain size, we suggest a two panel plot showing uranium concentration vs. grain size for a constant age and uranium concentration vs. age for a 63 micrometer grain size, with fractional [He] contamination as a heat mapped primary variable in each. We have attached examples.

We will modify the offending sentence in the conclusion to more closely follow the previous discussion.

**Response to Reviewer 2**

Cox et al. present in this manuscript a well-designed suite of experiments aiming to understand and quantify the trapping process of atmospheric helium at the surface of silicates during laboratory crushing. Crushing silicates (sometimes before melting them in vacuo) is a technique used in several labs for measuring helium isotopes and abundances (see for example Kurz et al., GCA, 1986; review in Blard, Chem. Geol., 2021). Since an unexpected atmospheric helium contamination may modify the geological signature of the samples, it is key to understand the physical process of this helium adsorption/trapping. This knowledge may improve the accuracy of future measurements.

Authors provide experimental observations confirming the previous conclusions of Protin et al. (GCA 2016), who initially reported this nearly-irreversible helium contamination. Cox et al. also identified mechanisms that favor (or limit) the adsorption of helium. They notably tested several procedures that may reduce this contamination, such as crushing in a liquid or a N2 pure atmosphere. They also developed a - quite (too?) complex - model to quantify the impact of this contamination on (U-Th)/4He thermochronometry.

In summary, I think this follow-up study is useful, because it brings new data and because it improves our knowledge of this strange mechanism that had unfortunately been overlooked for a long time. In my view, it thus deserves to be published. However, I noticed several points that also need to be considered in a revised version of the manuscript, before it can be published in its final form. All of them are quite straightforward and easy to address.

Main concerns

1 – Helium contamination when samples are crushed in vacuo before being exposed to atmosphere

Author make a strong case about the fact that crushing in vacuo leads to undetectable helium contamination. I suggest to nuance this statement in different places of the manuscript. Here is why: it is true that based on authors' data, crushing in vacuo led to almost undetectable contamination. However, their protocol is not very sensitive and do not permit to detect small amounts (< 10-13 mol/g) of helium contamination. Moreover, some data (Blard et al., EPSL, 2006) showed that crushing olivines/pyroxenes in vacuo followed by exposition to atmosphere may lead to detectable amounts (in the range of 10-14 to 10-13 mol/g) of adsorbed atmospheric 4He. These isotopic data have been reinterpreted later in (Protin et al., GCA, 2016) and (Blard, CG, 2021) using a mixing equation between atmospheric and magmatic helium. In the line of Cox et al's results, the amplitude of this "post vacuum crushing" contamination is indeed much lower than the one occurring when the crushing is performed in a He-rich environment. However, even such a low amount of adsorbed helium can be an issue that yields inaccuracies, particularly in the case of cosmogenic 3He measurements in samples having a low 3Hecosmogenic/4Hemagmatic ratio (see for example discussion in Blard, CG, 2021).

We agree that crushing in vacuum is not the best method for avoiding He contamination, according to both our results and the previous data referenced here. We will add some clarifying language to better emphasize this reality. However, we disagree with the assertion that our experiments are less sensitive to contamination. While the measurements of the Blard and Protin work is certainly far more sensitive, the method of He exposure we used (pure He rather than 5 ppm He in air) compensates for the difference. And in fact, we also see evidence of contamination in samples crushed in vacuum and then exposed to helium, which is part of the reason we recommend crushing in a neutral atmosphere instead.

2 – Complexity of the model used to compute the amount and impact of this contamination

I maybe missed the point, but the model described in section 4.1 is not trivial and I am wondering if such complexity is necessary. I can be wrong, it is just a feeling based on the often right "simpler is better" motto. Wouldn't you get similar results by simply deriving an empirical relationship between the amount of He-contamination and the specific surface area of the samples (as Protin et al., 2016 did)?

Why assuming an exponential vs depth decrease and what is the gain of this assumption regarding the accuracy of the model? Is this exponential attenuation more accurate than simply assuming that n atoms of He can be adsorbed per grain surface unit? Moreover, it is for me not clear how you compute the attenuation length of 0.29 micron-1, especially because you provide this number before describing the "onion shell" discrete model.

Equations should be numbered and better explained and quoted in the text, the rational would be easier to follow.

There are two separate points to address here. One is whether the model is overly complex. While it is true, at least for the simple spherical model we use here, that it would be simpler to just use a surface-area-to-volume relationship, our model has three major strengths over this approach. One, it attempts to describe the physical mechanism involved in the irreversible adsorption and provides a reasonable quantitative framework (the exponential decay of the trapping area away from the surface). At the same time, we emphasize that this is a hypothesis, and the model formulation here allows it to easily be substitute for a different relationship. Two, the model makes it trivial to introduce more complicated geometries including crystal structures and mixtures of multiple grain sizes. This seems valuable since the mechanism under discussion is of potential concern for a wide variety of measurements involving different noble gases, minerals, and sampling and processing methods. And third, the model allows us to account for the observation that most (but not all) of the trapped helium can be removed through chemical leaching of the outer several micrometers crystals.

The other point is whether the exponential relationship between trapping volume and depth is appropriate. This is just a hypothesis, but we think it is valuable to hypothesize a reasonable mechanism and then to produce a model that is consistent with it. This is again relevant to the new observation about the effect of chemical leaching in this study, and it would also be relevant to any attempt to use physical abrasion to treat contaminated samples

(one would also have to either eliminate He from the treatment atmosphere or determine if physical abrasion was sufficiently damaging to induce further cracking and trapping). I propose to expand on the model description to 1) emphasize the hypothetical nature of this physical model and 2) better explain the attenuation lengthscale we use, which is simply a fit of the chemical leaching data we have for a limited grain size range to the hypothesized function, and which is therefore independent of the onion shell model, which is a simple way to implement the function while preserving the ability to easily modify both the trapping volume vs. depth relationship and the crystal geometry. We will also take up the suggestion to number the equations and cite them in the text rather than placing them in line with the text as we did in the initial draft.

Other comments

Line 14: "Low temperature" is a rather imprecise and subjective statement. Provide actual temperature number.

We will change this to "room temperature." While we do discuss the fact that He is retained even under heating to temperatures as high as 800-900 degrees, if the process were reversible, He would be released again at room temperature after trapping at room temperature. "Room temperature" is also not very precise, but we did not record the exact temperature of the vacuum chamber.

Line 16: "…that this is easily avoided by crushing under liquid or in an inert atmosphere." I suggest to revise and nuance this sentence. Indeed, as I wrote above, some data (Blard et al., EPSL, 2006) suggest that crushing in vacuo followed by an exposition to atmosphere may also be subject to atmospheric helium adsorption/contamination (see data re-interpretation in Protin et al., GCA, 2016 and Blard, CG, 2021).

We show similar results for crushing in vacuum, but do not see the same effect under liquid or in an inert atmosphere.

Line 17: "samples to sizes smaller than typically used for geochronology". Provide grain sizes ranges in microns or mm.

Grain size will be stated (63 micrometers, or sand-sized, a common sieve size used for individual crystal picking).

Line 23: Quote few examples of relevant literature.

This is a good idea and we will provide example citations for each of the applications we list.

Line 23: Is "common" a synonym of "most abundant" in English? If not, you should prefer "most abundant".

These are synonyms, but we agree that "most abundant" sounds more technically correct and are happy to make the change.

Line 26: be more specific than "long-lived".

"Long-lived" in terms of isotope geochemistry is a bit of a moving target and is process- and measurement-dependent, but in this case, whatever cutoff one might choose is quite far from the 800 ms half-life of helium-6. We propose to change this to, "There are no radioactive isotopes of He long-lived enough to be found in measurable quantities in nature."

Line 27: "always" is a bit strong. I suggest "often".

We would argue that "always" is correct given that we state "relative to the number of potential components," but we suggest that we just delete "always" here.

Line 45: As written, this sentence may lead to think that "atmosphere" and "radiogenic" endmembers could have similar 3He/4He isotopic ratios (which is not the case at all). I would rephrase. The main limitation is that He doesn't have a third isotope. If this was the case, it could permit to resolve a 3 components mixing.

This is not the only problem, though, since the 3He/4He of radiogenic helium varies considerably. We concede that simply calling this quantity "uncertain" is misleading. We suggest changing to, "due to the extremely low and relatively uncertain 3He/4He ratio of air and the low and variable 3He/4He ratio of radiogenic He."

Line 46: I suggest "mixture" instead of "contamination" here. In my opinion, contamination should be reserved to unexpected addition of helium during experimental procedures.

Fair point--we will make the change.

Line 48: It would be informative to quote here the most recent - high precision - survey published by (Boucher et al., GCA, 2018). Analytical precision of the 3He/4He ratio is close to 1 per mil with this apparatus.

We will add this citation to the first part of this statement, although it doesn't change the point since this is not an absolute determination.

Lines 52-55: I think this statement should be nuanced, and you could recall here that the nucleogenic 3He contribution may be significant for the accuracy of cosmogenic 3He determinations, in minerals having pluri-million years He-closure ages (see for example figure 7 in Blard, CG, 2021).

We will add a brief passage after this one mentioning this work and more completely addressing cosmogenic measurements as well.

Line 63: About the choice of the "best" granulometric window, you could also mention here the case of cosmogenic 3He analysis. (Blard, CG, 2021) proposes that the optimal granulometric window is probably 100-500 microns. Grains larger than 500 microns should be avoided since they bear large amount of magmatic helium, while fractions finer than 100 microns can be contaminated by too much atmospheric helium.

Again, we appreciate the suggestion to better address cosmogenic measurements, and we will take up this recommendation.

Lines 73 to 79: You should mention here two other important reasons that have often motivated the measurement of crushed fractions in previous studies devoted to cosmogenic 3He determinations: 1) crushing (followed by fusion of the obtained powdered phenocrysts) is the standard procedure proposed by Kurz et al., GCA 1986 to analyze the magmatic 3He/4He endmember. 2) Several studies (Puchol et al. CG 2017; Blard CG 2021) also showed that small grains (< 500 microns) bear less fluid inclusions and hence much less amount of magmatic helium, leading to lower magmatic corrections, and lower uncertainty associated with the final cosmogenic 3He concentration (see Figure 5 in Blard, CG, 2021).

We will add these.

Line 95: Can you provide the range of this magmatic He estimates?

We think providing a range (and implying some sort of statistical confidence in it) from this limited set of measurements would be more misleading than not, so we suggest to simply eliminate "and sometimes much less."

Line 95 (and everywhere else where this reference appears): Remove the "." after "Murty.".

This will be corrected.

Line 96: "highest", "small". These subjective words are not very informative, especially for non-expert readers. Provide real numbers.

We will change this to reference the value in the previous sentence and to provide a calculation for the smallest amount of He contamination in a 50-125 um sample (sample c) provided in Protin et al. (2016), which is .24e11 atoms, or 4e-14 moles, in 0.4853 grams, or 8.2e-14 mol/g contamination from atmospheric helium. Scaling from 5 ppm to pure He, this would be 200,000 times higher or 1.6e-8 mol/g.

Line 133: What is the blank/signal ratio (in %) and what is the uncertainty associated with this blank?

We will add uncertainties to the numbers here and state blank/signal, which ranges from 8% to 73% on first extractions.

Figure 1: I suggest to find a way to indicate on the figure that the dark green box represents the 1500°C extraction (and light green 800°C). Why the N2 experiment yields 4He concentrations that are lower than the background? Do you suspect magmatic helium loss during the crushing? In that case, the "background" you use is a clear overestimate. Could you compute the 2 endmembers budget (magmatic He + atmospheric adsorbed He) by using the isotopic ratios measured with the MAP?

We have made modifications in response to this comment and to reviewer 1. The new version is attached. Regarding the N2 experiment, this difference represents instead the range of low initial He concentrations in the parent material, for which we do not have good isotopic information. We used pure He and He concentrations to avoid having to account for this--we are only concerned with significant He contamination here, not with the isotopic composition of He in San Carlos olivine.

Lines 164-165: I suggest to nuance this statement here and elsewhere. Indeed, some data (Blard et al., EPSL, 2006) showed that crushing olivines/pyroxenes in vacuo followed by exposition to the atmosphere may lead to detectable adsorption of atmospheric helium. These data have been reinterpreted in Protin et al., 2016 and Blard, 2021. Yes, the amplitude of this contamination is much lower than the one occurring when crushing is performed in a He-rich environment.

This kind of surface adsorption is distinct from the kind of irreversible trapping we are discussing here, and likely accounts for the exposure-after-vacuum contamination we see as well. We will address this work further as described above.

Line 167: I propose that "virtually no additional" should rather be "non-detectable adsorption in excess to the magmatic helium background of the samples". Since the uncertainty associated with this background may be large, maybe your experiment does not permit to detect the adsorbed helium (notably in the case of the N2 crushing experiment).

The use of pure He makes our experiment extremely sensitive to small amounts of contamination. The uncertainty associated with the background is not large compared to the degree of contamination observed in samples crushed in the presence of pure He.

Line 192: Quote here (Niederman and Eugster, GCA, 1992), who were the first to evidence the adsorption of xenon and krypton on silicate surfaces.

We will include a citation to this work.

Lines 193-194: The words "shortly" and "minimal" should be revised. As written above, previous data (Blard et al., 2006; reinterpreted in Protin et al., 2016 and Blard 2021) show that exposing phenocrysts to the atmosphere (24 to 48 hours after they had been crushed in vacuo) may induce a [10-14 – 10-13] mol/g contamination by atmospheric helium.

We show that the degree of contamination is not zero, but is dramatically smaller after a brief delay, so we think the language we use is appropriate here. Our goal is to show when the observed effects are important and the best way to avoid analytical problems for most measurements, but we will add more nuanced discussion of this previous work as described above.

Lines 201-202: Nuance this statement. Atmospheric helium can also be adsorbed and detected after in vacuo crushing (Blard et al., EPSL, 2006; revised interpretation in Protin et al., 2016 and Blard, 2021). The amplitude of the contamination is lower in the case of in vacuo crushing, but it may remain an issue in some cases, for example for cosmogenic 3He determinations that require very accurate and precise magmatic 3He estimates (case of samples having a low 3Hecosmogenic/4Hemagmatic ratio).

We already begin this paragraph by acknowledging that there are some cases in which the problem remains an issue, and we do suggest crushing in an neutral atmosphere rather than in vacuum based on these and our own results, but we will add a note also stating that low cosmogenic 3He samples may require special consideration.

Lines 204-206: Even if this statement is true for samples being very rich in radiogenic 4He (> 10-11 mol.g-1), it is much less true in the case of samples used for cosmogenic 3He measurement. In the dataset presented by (Blard et al., 2016), the <140 microns of in vacuo crushed samples were affected by magmatic 3He overcorrection representing from 10 to 100% of the cosmogenic 3He concentration.

See above.

Line 209: I don't understand the justification of using the adverb "only" here. On the contrary, 90% of He removal seems to be a significant proportion.

We think the next sentence already explains what we mean here. 90% is a lot but not enough, and not as good as simply crushing in a neutral atmosphere. We will, however, add more discussion of the implicaitons of this experiment as we also better describe the exponential contamination model.

Line 217: It would be very useful if you could provide a number to characterize the detection limit of your methodology. I guess this is the average of the magmatic 4He concentrations of the San Carlos olivine, plus 3 times its associated uncertainty? How well do we know these two values?

This is explained in lines 98-101, but we will reference it again here.

Lines 220-225: Here again it would be useful to quote (Niederman and Eugster, GCA, 1992), since they explored these mechanisms for heavy noble gases.

We will add this reference.

Section 4.1 – Model description. If you wish (and have enough energy and time!) to add half a page and one figure, I suppose that the same modeling approach applied to cosmogenic 3He dating would be great (considering the impact of the atmospheric contamination on the magmatic 3He correction). If you stay with the evaluation of the impact of this contamination to (U-Th)/4He thermochronology only, then replace the word "geochronology" (line 229) by "thermochronology".

This is a good idea, and we are happy to add this in addition to the additions requested by the other reviewer. Combined with the response to Reviewer 1, this would mean that Figure 3 becomes a 4 panel figure, with two new panels created for cosmogenic 3He as well. We are open to additional suggestions about how best to make this figure useful for cosmogenic users.

Line 233: Are they any experimental or theoretical basis justifying the choice of an exponential law to describe the vertical attenuation of the trapping?

This is a hypothesis. We will add a brief section better explaining the origin of the number, which is based on the leaching experiment we performed.

Lines 237 to 239: In the case of cosmogenic 3He measurements, atmospheric 4He adsorption is the biggest potential issue: if overlooked, this may induce an overestimate of the magmatic 3He correction, and an underestimate of the cosmogenic 3He (see discussions in Protin et al., 2016 and Blard 2021).

We will add a statement about this issue as well.

Line 249: Add "microns" after "22.5".

This oversight will be corrected.

Line 256: So, a = R(n) - R(n-1)?

Yes, we can add this equation for additional clarity.

Line 263: "total concentration" should be replaced by "total contamination", because the magmatic and matrix-sited Helium species are not considered here (I think).

Both are correct (because the other He is ignored), so we will clarify here that all of the He considered represents "contamination."

Line 266: "...for He measurements in thermochronology". In the case of cosmogenic 3He determinations, the most commonly used minerals are olivines and pyroxenes.

We will add this caveat.

Line 295: Suggestion: "… the geometric model."

We will make this change.

Line 298: Suggestion: "need to be".

"Need" is semi-modal, so both are correct, but we are happy to make the change.